# Textile Composite Damage Analysis Taking into Account the Forming Process

**DOI:** 10.3390/ma13235337

**Published:** 2020-11-25

**Authors:** Marjorie Jauffret, Aldo Cocchi, Naim Naouar, Christian Hochard, Philippe Boisse

**Affiliations:** 1INSA-Lyon, LaMCoS CNRS, University de Lyon, 69621 Lyon, France; marjorie.jauffret@insa-lyon.fr (M.J.); naim.naouar@insa-lyon.fr (N.N.); 2LMA CNRS, Aix Marseille University, 13007 Marseille, France; cocchi@lma.cnrs-mrs.fr (A.C.); hochard@lma.cnrs-mrs.fr (C.H.)

**Keywords:** damage, forming, hypoelasticity, textile composite, fibre orientation

## Abstract

The internal structure of composite materials is modified during manufacturing. The formation of woven prepregs or dry preforms changes the angle between the warp and weft yarns. The damage behaviour of the consolidated composite is modified by these changes of angle. It is important when designing a composite part to consider this modification when calculating the damage in order to achieve a correct dimensioning. In this paper, a damage calculation approach of the consolidated textile composite that takes into account the change in orientation of the yarns due to forming is proposed. The angles after forming are determined by a simulation of the draping based on a hypoelastic behaviour of the woven fabric reinforcement. Two orthogonal frames based on the warp and weft directions of the textile reinforcement are used for the objective integration of stresses. Damage analysis of the cured woven composite with non-perpendicular warp and weft directions is achieved by replacing it with two equivalent Unidirectional (UD) plies representing the yarn directions. For each ply, a model based on Continuum Damage Mechanics (CDM) describes the progressive damage. Two examples are presented, a bias extension specimen and the hemispherical forming coupon. In both cases, the angles between the warp and weft yarns are changed. It is shown that the damage calculated by taking into account these angle changes is greatly modified.

## 1. Introduction

When the ratio between stiffness and weight is important in the design of a structure, composite materials are a preferred solution. This is particularly the case for many aeronautical structures [1,2] as well as in the automotive industry [3,4,5]. The reinforcements and matrix can be adapted to a given application. However, the manufacturing process often changes the final state of the composite, especially when forming is performed. The directions of the reinforcement fibres are modified, both in the case of Liquid Composite Moulding processes (LCM) [6,7] as well as in the thermoforming of thermoset or thermoplastic prepregs [4,8,9]. The simulation of the forming process of composite parts has developed significantly in recent years [10,11]. These advancements make it possible to predict the state of the composite after manufacture in particular the orientation of the fibres and certain defects such as wrinkles [12,13,14]. Some studies have been conducted on the influence of the manufacturing processes on the mechanical characteristics of composites [15,16,17]. The influence of defects and in particular of wrinkles and waviness on the mechanical characteristics of composites were analysed in [18,19,20,21].

Damage is one of the major degradations of composite parts during service. Micro-cracking of the matrix can lead to failure of the composite. Damage analysis and modelling is an important aspect of composite science, especially since damage modes can be complex due to the fibre-matrix composition of composites. This paper analyses the influence of angle changes between warp and weft yarns created by the forming of the textile reinforcement on the development of damage in the consolidated composite. The simulation of the forming of a woven reinforcement is based on a hypoelastic approach. One of the main results of this simulation is the prediction of the fibre orientations after the forming process. A damage model for reinforced woven-composites is presented for the analysis of the consolidated composite. This model is based on Continuum Damage Mechanics (CDM) and can describe the progressive damage evolution of the material which consists in the propagation of small matrix cracks parallel to the fibre directions. The model, initially proposed for unidirectional plies [22], has been extended to woven plies [23,24]. The woven ply is modelled by two equivalent UD plies that correspond to the warp and the weft directions, directions that may be non-orthogonal after the forming process. After this simplification, it is possible to directly apply the classical unidirectional model [22] that describes the linear elastic behaviour in the fibre direction and the nonlinear damaged behaviour in the transverse and shear directions [23].

To our knowledge, there is a lack of simulation techniques suitable for the estimation of matrix damage in a loaded composite structure accounting for the effect of the forming process. This represents the objective of the present article in the case of textile composites. It will also be shown that this consideration of the forming process has important consequences on the calculated damage. To demonstrate the potential of coupling the two approaches of forming simulation and damage calculation, first a bias extension test (in-plane shear test obtained by traction at 45° [25,26,27]) at high temperature on a thermoplastic prepreg with a woven reinforcement is performed and simulated. Afterwards, a tensile test is performed on the consolidated composite with modified warp and weft angles due to the forming process. The damage generated in the coupon for this configuration is then simulated and analysed. In the last part, a tension test on a coupon obtained by hemispherical forming is simulated. It is shown that, in this case, the damage level in zones where the angle variations between warp and weft are large is strongly affected by the forming process.

## 2. Forming Process Simulation

The simulation of composite forming has been the subject of much work in recent decades. Reviews in this domain are presented in [10,11]. In LCM processes, the textile reinforcements are dry (resin-free) during the forming process, which takes place before the resin is injected. The thermoforming of prepregs is carried out at a temperature such that the matrix is melted to allow the forming. The presence of the resin in this case conditions the mechanical behaviour of the composite during forming by making it dependent on temperature and giving it a viscous character [28,29,30,31]. In all cases, the mechanical behaviour during forming is mainly driven by the reinforcement and the position of the fibres, which are quasi inextensible. Concerning simulation techniques, one can distinguish between kinematic methods [32,33,34,35] which are numerically fast but do not take into account the nature and the forces acting on the reinforcement such as those generated by a blank-holder, and approaches based on the equations of mechanics, the behaviour of the reinforcement and the frictional contact between the blank and the tools. Because the textile reinforcements are thin, some approaches consider them as membrane where the bending stiffness is considered low enough to be neglected [36,37,38,39,40,41]. It has nevertheless been shown that the bending stiffness of the textile reinforcement plays an important role in particular on the development of wrinkles [14,42,43]. In this article, the objective of the forming simulation is mainly the determination of warp and weft yarns angle variation, which is related to the membrane deformation of the textile reinforcement. Therefore, a membrane approach will be used. The textile reinforcement is assumed to be continuous and a hypoelastic behaviour model is used. Hyperelastic approaches [44,45,46] and elastic approaches [11,47,48,49] have also been developed.

### 2.1. An Hypoelastic Approach Based on the Warp and Weft Yarn Rotations

Hypoelastic behaviour models (also known as ‘rate constitutive equations’ [50]) have been developed to model the mechanical behaviour of materials under large deformations. These behaviour laws are frequently used and are often proposed by default in finite element codes. The constitutive law presented below is implemented in the explicit Abaqus code via a Vumat user subroutine. From the strain rate $\underline{\underline{D}}$, a constitutive tensor $\underline{\underline{\underline{\underline{C}}}}$ gives the objective derivative of the Cauchy stress tensor. This is the strain rate for an observer fixed to the material.
(1)$${\underline{\underline{\sigma }}}^{\nabla }=\underline{\underline{\underline{\underline{C}}}}:\;\underline{\underline{D}}$$
with
(2)$${\underline{\underline{\sigma }}}^{\nabla }=\underline{\underline{\overset{•}{\sigma }}}+\underline{\underline{\sigma }}.\underline{\underline{\Omega }}-\underline{\underline{\Omega }}.\underline{\underline{\sigma }}$$

Here, $\underline{\underline{\overset{•}{\sigma }}}=\frac{d\sigma }{dt}$ is the time derivative of the Cauchy stress. The spin $\underline{\underline{\Omega }}=\underline{\underline{\overset{•}{Q}}}.{\underline{\underline{Q}}}^{T}$ is associated to the rotation $\underline{\underline{Q}}$ of the frame that rotates with the material. The use of the objective derivative of stresses is necessary to ensure that movements of rigid bodies do not create stresses. There are several objective derivatives. The most classical are Green Naghdi and Jaumann [51,52]. It has been shown that these derivatives are not suitable for fibrous materials. In this case, the objective derivative must be that defined by the rotation of the fibre [53,54]. The directions ${\underline{f}}_{1}$ and ${\underline{f}}_{2}$ of warp and weft yarns during deformation are given by the initial positions of the yarns ${\underline{f}}_{10}$ and ${\underline{f}}_{20}$ and the deformation gradient $\underline{\underline{F}}$.
(3)$${\underline{f}}_{1}=\frac{\underline{\underline{F}}.{\underline{f}}_{10}}{\left\|\underline{\underline{F}}.{\underline{f}}_{10}\right\|}\;{\underline{f}}_{2}=\frac{\underline{\underline{F}}.{\underline{f}}_{20}}{\left\|\underline{\underline{F}}.{\underline{f}}_{20}\right\|}$$

Two orthogonal frames based on these two yarn directions are defined: g **(g_1 =_f_1_, g_2_)** and h **(h_1_, h_2 =_f_2_****)** (Figure 1).

### 2.2. Stress Update in the Frames of the Yarns

The strain increment between times t^n^ and t^n+1^ is denoted $\underline{\underline{d\varepsilon }}$. Its components in the frames g **(g_1_, g_2_)** and h **(h_1_, h_2_****)** are considered (α and β are indexes taking the value 1 or 2).
(4)$$\underline{\underline{d\varepsilon }}=d\varepsilon_{\alpha \beta }^{g}{\underline{g}}_{\alpha }⊗{\underline{g}}_{\beta }=d\varepsilon_{\alpha \beta }^{h}{\underline{h}}_{\alpha }⊗{\underline{h}}_{\beta }$$

The incremental elongation in the warp yarn direction ${\underline{g}}_{1}$ gives the incremental axial stress component in ${\underline{g}}_{1}$ direction from tensile stiffness E^g^.
(5)$$d\varepsilon_{11}^{g}={\underline{g}}_{1}.\;\underline{\underline{d\varepsilon }}\;.\;{\underline{g}}_{1}\;d\sigma_{11}^{g}=E^{g}d\varepsilon_{11}^{g}\;$$

In the same way, the stress increment in the weft direction ${\underline{g}}_{2}$ is obtained by:(6)$$d\varepsilon_{22}^{h}={\underline{h}}_{2}.\;\underline{\underline{d\varepsilon }}\;.\;{\underline{h}}_{2}\;d\sigma_{22}^{h}=E^{h}d\varepsilon_{22}^{h}\;$$

The in-plain shear stress increment is obtained from the in- plain shear strain increment and the shear modulus G.
(7)$$d\varepsilon_{12}^{g}={\underline{g}}_{1}.\underline{\underline{d\varepsilon }}.{\underline{g}}_{2}\;\;d\varepsilon_{12}^{h}={\underline{h}}_{1}.\underline{\underline{d\varepsilon }}.{\underline{h}}_{2}\;d\sigma_{12}^{g}=Gd\varepsilon_{12}^{g}\;\;d\sigma_{12}^{h}=Gd\varepsilon_{12}^{h}$$

The shear modulus G is not constant and varies significantly with the shear strain [25,26,27]. The stress components are integrated between t^n^ and t^n+1^ following the scheme of Hughes and Winget [49] in g **(g_1_, g_2_)** and h **(h_1_, h_2_****)** frames, respectively.
(8)$${\left(\sigma_{11}^{g}\right)}^{n+1}={\left(\sigma_{11}^{g}\right)}^{n}+d\sigma_{11}^{g^{n+1/2}}\;{\left(\sigma_{12}^{g}\right)}^{n+1}={\left(\sigma_{11}^{g}\right)}^{n}+d\sigma_{12}^{g^{n+1/2}}\;\;{\left(\sigma_{22}^{h}\right)}^{n+1}={\left(\sigma_{22}^{h}\right)}^{n}+d\sigma_{22}^{h^{n+1/2}}\;{\left(\sigma_{12}^{h}\right)}^{n+1}={\left(\sigma_{11}^{h}\right)}^{n}+d\sigma_{12}^{h^{n+1/2}}\;$$

The Cauchy stress tensor at time t^n+1^ is then determined by: (9)$${\underline{\underline{\sigma }}}^{n+1}={\left({\underline{\underline{\sigma }}}_{}^{g}\right)}^{n+1}+{\left({\underline{\underline{\sigma }}}_{}^{h}\right)}^{n+1}\;$$

The components of this stress tensor can be computed in any frame. In particular, the Abaqus explicit solver requires that the Vumat user routine gives the components in the Green–Naghdi’s frame that rotates with the polar rotation.

In addition to the stresses, the approach described above provides the directions ${\underline{f}}_{1}$ and ${\underline{f}}_{2}$ of warp and weft yarns (Equation (3)). These directions, and thus the angle change between warp and weft yarns, are considered in the damage calculation of the consolidated composite presented in the next sections. The following two tests will be considered to analyse the influence of the forming and more particularly of the warp-weft angle change in the calculation of the damage of the consolidated composite. The Young modulus E^h^ and E^g^ of Equations (5) and (6) (in warp and weft directions) and the shear modulus G of Equation (7) are given in Table 1 for the woven reinforcement considered in the simulations in Section 2.2 and Section 2.3.

### 2.3. Bias Extension Test 

The Bias Extension Test is an experiment fairly simple to perform that imposes in-plane shearing on a specimen with a textile reinforcement [25,26,27]. Warp and weft yarns must be initially oriented at ±45°. The specimen is extended creating constant shear zones (A,B,C) in Figure 2. For the test to be carried out, the length of the specimen must be greater than or equal to twice its width. For the specimen utilised here, the dimensions are 210 × 70 mm^2^. The prepreg utilised is a PA66 thermoplastic matrix reinforced with an 8-Harness Satin glass woven. The bias extension test can be carried out either on dry textile reinforcements (without resin) or on prepregs at a sufficiently high temperature for the resin to be molten. This last technique is applied in our case. The test is performed at 270 °C, i.e., slightly above the melting temperature of the PA66 matrix. The deformation of the specimen is obtained by a 50 mm displacement of the moving jaw of the traction machine (d in Figure 2a) which corresponds to a warp and weft angle of 38°. The specimen is left cooling obtaining the final configuration where the warp and weft yarn directions have been modified by the bias extension.

### 2.4. Hemispherical Forming

The simulation of a hemispherical forming is then performed (Figure 3).

The hemispherical shape is widely used to perform draping analysis for all materials [55,56] but especially for textile reinforcements of composites [12,44,57]. It is simple in shape but clearly double curved and requires large plane shear deformations for its shape to be achieved [57,58,59,60,61,62]. This forming process leads to angles between warp and weft yarns, which evolve with the position between 0° and 45° (Figure 4). This will allow the influence of a variable angle between warp and weft to be analysed below.

The two tests presented in this section, bias extension test and hemispherical forming, both strongly modify the angles between warp and weft yarns. The objective of the following sections is to assess the influence of these modifications on the damage behaviour of the consolidated composite.

## 3. Damage Model for Woven Ply Laminates

This section concerns the cured composite after the manufacturing process. The failure of laminate composite structures is due to many mechanisms acting at different scales and depending on the type of ply, woven or unidirectional (UD), concern the matrix or the fibres and are a function of many parameters (thickness, orientations, loads, etc.) [63,64,65,66,67,68,69,70]. Matrix damage usually starts with fibre/matrix decohesions that propagate in to form transverse cracks over the entire thickness of the ply. In the case of UD plies, these transverse cracks can spread over great lengths and can initiate delamination [71,72,73,74]. In the case of woven plies, these transverse cracks develop over the whole thickness of the yarns, but their propagation is blocked by the crossing of the warp and weft fibres. The models based on Continuum Damage Mechanics (CDM) [22] can described the progressive damage of the material. These CDM models are suitable for woven composites because matrix cracks are distributed more regularly than for UD plies [23]. Also, laminates made of woven plies exhibit better resistance to delamination. It is therefore possible to remain within the framework of the theory of laminates in plane stresses and CDM to describe the evolution of matrix damage for static and fatigue loads [75,76].

A simplified model has been proposed to describe the behaviour of woven plies within the framework of 2D laminate theory. In order to describe the different evolutions of matrix damage in the warp and weft, the woven ply is replaced by two Equivalent UD (EqUD) plies whose thicknesses respect the ratios between the number of warp and weft yarns. In the case of a balanced woven ply, the thicknesses of the EqUD plies are the same. In the case of unbalanced woven the thicknesses of the EqUD plies will be different. This modelling strategy allows for the simulation of woven reinforcement where the warp and weft yarns are non-perpendicular, this way the orientations of the yarns can correspond to the ones obtained after forming; it is this original point that will be used in this paper. After this simplification, it is possible to directly apply classical UD models, [22] for example, describing the linear elastic behaviour in the fibre direction and the damaged non-linear matrix behaviour in the transverse and shear directions.

### 3.1. Equivalent Unidirectional Plies Model of the Woven Ply

The woven ply is replaced by two Equivalent Unidirectional (EqUD) plies with different thicknesses to take into account the different ratios of fibres in the warp and the weft directions (Figure 5). This model applies to both balanced and unbalanced woven fabrics because it takes into account a parameter δ which represents the percentage of fibres in the warp direction, 1-δ therefore being the percentage in the weft direction. In this study, we consider only balanced woven plies, so δ will be always equal to 0.5.

The elastic parameters and the damage law coefficients of the EqUD plies were evaluated from the mechanical response of the woven ply laminates. The identification of the material properties required by the model is detailed in [23,77] for an unbalanced woven glass/epoxy plies and static loads. The elastic and damage laws have been tested and validated for balanced woven carbon/epoxy plies in static [76,78] and extended for fatigue loads [75,76]. The proposed simplified model of using two equivalent UD plies to model the behaviour of a woven ply has been used recently by other authors [79,80,81]. 

### 3.2. Matrix Damage Model at the Unidirectional Ply Scale

#### 3.2.1. Assumptions

A model based on Continuum Damage Mechanics (CDM) has been developed to describe the damage evolution in composite materials at the ply scale [22]. 

Material damage can be described by its effects on elastic coefficients [82,83]. In the case of a ply consisting of continuous fibres bound by a brittle epoxy matrix, matrix cracks parallel to the direction of the fibres will appear very early, spreading rapidly throughout the thickness of the ply and also in the plane (Figure 5a). The density of these cracks will gradually increase with loading and lead to a decrease in transverse and shear moduli at the ply scale. While the development of these matrix cracks is gradual and generally not catastrophic for the laminate, the evolution of fibre cracks is much more abrupt. In the context of CDM, it is conventional to introduce internal damage variables *d_i_* to describe these decreases:(10)$$E_{1}^{}=E_{1}^{0}(1-d_{1}^{})\;;\;E_{2}^{}=E_{2}^{0}(1-d_{2}^{})\;;\;G_{12}^{}=G_{12}^{0}(1-d_{12}^{})$$
where $E_{1}^{0},\;E_{2}^{0}\;\mathrm{and}\;G_{12}^{0}$ are the initial moduli and the damage variables *d*_1_, *d*_2_, *d*_12_ vary from 0 for the initial material to 1 for the fully degraded material:-*d*_1_ whose evolution represents the linear elastic behaviour and the brittle fracture of the fibres observed during a tensile test in the longitudinal direction of the fibres;-*d*_2_ representing the effect of the matrix damage on the stiffness in the transverse direction;-*d*_12_ representing the effect of the matrix damage on the shear stiffness.

The damage is assumed to be uniform in the thickness of the ply.

The progressive development of the damage variables *d*_2_ and *d*_12_ depends on the tensile and shear stresses, which generate cracks in the matrix. Under the assumption of plane stress and small perturbations, the strain energy in the ply can be written as follows [22,23]: (11)$$E_{D}^{ps}=\frac{1}{2}\left[\frac{{\langle \sigma_{1}\rangle }_{+}^{2}}{E_{1}^{0}(1-d_{1})}+\frac{{\langle \sigma_{1}\rangle }_{-}^{2}}{E_{1}^{0}}+\frac{{\langle \sigma_{2}\rangle }_{+}^{2}}{E_{2}^{0}\left(1-d_{2}\right)}+\frac{{\langle \sigma_{2}\rangle }_{-}^{2}}{E_{2}^{0}}-2\frac{\nu_{12}^{0}}{E_{1}^{0}}\sigma_{1}\sigma_{2}+\frac{\sigma_{12}^{2}}{G_{12}^{0}\left(1-d_{12}\right)}\right]$$
where < . >+ and < . >- are the MacAuley brackets which select respectively the positive or the negative part of the quantity that is within. This way, the tensile energy and the compressive energy are separated in order to describe the unilateral nature of the damage progression given by the opening of the cracks in tension and their closing in compression. The thermodynamic forces associated with the internal tensile and shear variables *d*_1_, *d*_2_ and *d*_12_ are defined as follows:(12)$$\{\begin{array}{c} \begin{array}{l} Y_{d_{i}^{}}=\frac{\partial E_{D}^{ps}}{\partial d_{i}^{}}=\frac{{\langle \sigma_{i}\rangle }_{+}^{2}}{2E_{i}^{0}{\left(1-d_{i}^{}\right)}^{2}}\;\;with\;i=1,\;2 \end{array}\\ Y_{d_{12}^{}}=\frac{\partial E_{D}^{ps}}{\partial d_{12}^{}}=\frac{{\left(\sigma_{12}\right)}^{2}}{2G_{12}^{0}{\left(1-d_{12}^{}\right)}^{2}} \end{array}$$

#### 3.2.2. Damage Evolution Laws

The development of the internal damage variables depends on these thermodynamic forces and, more specifically, their maximum value during the loading history.

Under tensile load, *d*_1_ increases suddenly to model the brittleness behaviour in the direction of the fibres.

By noting $Y_{1}^{max}$ the resistance in the fibre direction, we then have: (13)$$d_{1}^{}=\dot{0\;\mathrm{if}\;}Y_{d_{1}^{}}^{}<Y_{1}^{max}\;;\;d_{1}^{}=\dot{1\;\mathrm{otherwise}}$$

In the case of static loading, the tension/shear coupling during the progressive development of *d*_2_ is accounted for by the following equivalent thermodynamic force: (14)$$Y_{eq}=a\;{\left(Y_{d_{2}^{}}\right)}^{n}+b\;{\left(Y_{d_{12}^{}}\right)}^{m}$$
where a, b, m and n are material parameters specifying the tension/shear coupling. The evolution law for the damage is written as:(15)$$d_{2}^{}={\langle 1-e^{-\left(Y_{eq}-Y_{0}^{s}\right)}\rangle }_{+}$$
where the constant $\;Y_{0}^{s}$ corresponds to the threshold value for the development of *d*_2_.

As proposed in [23,77], the shear damage *d*_12_ is taken proportional to the transverse damage *d*_2_. This choice is based on the fact that the cracks are parallel to the fibres and their effect on the transverse and shear modulus is the same.
(16)$$d_{12}=c\;d_{2}$$

By considering that the moduli are proportional to the effective surfaces (initial surfaces minus the cracked areas) [83], one can take as a first approximation the coefficient *c* = 1. This coefficient probably varies as a function of the crack density and numerical micromechanical analyses [81] showed that it is less than 1 but increases for high damage levels.

#### 3.2.3. Inelastic Strain in the Shear Direction

After loading has been applied to a [45,−45]s laminate for example, inelastic strains are observed. These strains may result from the slipping/friction process occurring between fibre and matrix as the result of the damage. Although inelastic strain can also be observed during tensile test on 0° and 90° laminates, only the inelastic strain in the shear direction is relevant. A kinematic hardening model was used to describe the inelastic shear strain evolution.

The coupling between the damage and the plasticity is accounted for by the effective stress and the effective strain [22,23] which are respectively written as:(17)$${\tilde{\sigma }}_{12}=\frac{\sigma_{12}}{\left(1-d_{12}\right)}\;\;\;\;\;\mathrm{and}\;\;\;\;\;{\tilde{\varepsilon }}_{12}^{p}=\varepsilon_{12}^{p}(1-d_{12})$$

It is assumed that stresses $\sigma_{11}$ and $\sigma_{22}$ do not influence the elastic field domain defined by: (18)$$f=\left|{\tilde{\sigma }}_{12}-h{\tilde{\varepsilon }}_{12}^{p}\right|-R_{0}$$

Corresponding to a kinematic linear hardening law where *R*_0_ is the initial inelastic strain threshold and *h* is the linear law coefficient. 

The identification of parameters of the damage model presented above are given in details in [23]. The identification of the behaviour typically requires three loading-unloading-reloading tests: a test at 0° for the behaviour in the fibre direction (Figure 5b) [84], a test at 90° for the transverse behaviour and a test on a [45,−45] s laminate to identify the behaviour in shear (Figure 5b). The numerical values of the parameters adopted in the simulations are given in the Table 2. The papers [23,84] present several validation tests on different laminates for static loads. The model has been extended to fatigue loads [75,76]. In this case, the matrix damage evolves during the cycles and the model has been identified and validated. It should be noted that for each type of fibre, resin or weaving, it is necessary to repeat these tests to identify the damage model coefficients of the equivalent UD ply.

The part of the model presented here relates mainly to the nonlinear evolution of the matrix damage for static loadings. Laminate failure is not discussed here. This more complex part is presented in the cited papers. Catastrophic laminate failure is rarely due to matrix damage localization mechanisms because the laminate usually consists of plies oriented in the loading direction. On the other hand, the fibre failure is catastrophic for the laminate and the structure. The approach used in this case is described in [85] and takes into account the resistance decrease in the fibre direction as a function of the level of matrix damage and also takes into account the increase in fibre strength in the presence of stress concentrations using non-local criteria [84,86]. It should be noted that these non-local criteria make it possible to regularize the problem and thus avoids the mesh-size dependence for the numerical approaches.

## 4. Damage Analysis Taking into Account the Yarns Angle Variations Due to Forming

### 4.1. Bias Extension Test

A tension test simulation was performed on the consolidated specimen obtained after a bias extension test. As described in Section 2.3, after forming the specimens presents different yarn orientations, the original [45,−45] s layup is conserved only at the extremities and a [19,−19] s orientation is found in the central zone. To perform the simulation the left end is fixed (Figure 6). The right end is subjected to a horizontal force of 7000 N and vertical displacement are blocked.

#### 4.1.1. Transverse Damage d_2_

Figure 6 presents the matrix transverse damage map. It can be observed that the damage is very low in the central zone (dark blue), where the orientation of the fibres is [19,−19] s with respect to the loading direction. In this area the orientation of the yarns is close to the loading direction, therefore, the transverse and shear stresses responsible for matrix damage are the lowest. In the zones at the ends of the test specimen, the yarn angle remains unchanged after forming and therefore close to the initial orientation of [45,−45] s. In these zones, where the loading direction is far from the orientation of the fibres, the damage reaches its maximum values in the order of 0.5, which corresponds to a loss of rigidity of 50% (in green) because the shear stresses at the scale of the EqUD plies are maximum. 

In this example, the damage shows a strong variation depending on the angles between warp and weft fibres and the loading direction. It should be noted that with the described damage model, two damage maps are obtained: one for the EqUD ply corresponding to the warp direction and another for the EqUD ply to the weft direction. As the applied force follows the bisector between the original warp and weft directions, the results obtained for the two EqUD plies are very similar for this example.

#### 4.1.2. Analysis of Inelastic Strains

For the studied case inelastic strains are present only in the zones of strong yarn angle variation especially at the meeting point between the A and B zones where the yarn orientation is [45,−45] s (Figure 7).

### 4.2. Damage Analysis on a Specimen from Hemispherical Forming

A rectangular specimen is extracted from the flat zone of the part shaped by hemispherical stamping. This specimen is subjected to tension and the damage is analysed. In this coupon, the angle between the yarns of the first ply varies gradually but strongly (Zone A, in Figure 8). The boundary conditions are the same as in the case of the bias extension test. The left end of the specimen is fixed, and the right end is subjected to a tension force while transverse displacements are blocked.

Figure 9 clearly shows that the direction of the fibres plays a major role in the development of damage when the specimen is subjected to tension. The left part where the warp and weft yarns have a direction quite close to that of the loading direction is little or not damaged. The damage starts and then becomes significant in the right side of the specimen where the weft and warp directions are close to [45,−45] s. This example shows that taking into account changes in orientation due to forming is essential in the damage analysis of composites structures obtained by forming techniques. We can see that the strongly nonlinear evolution of the damage which increases rapidly in the right part of the specimen when the force varies only from 4000 to 5000 N.

The inelastic strains associated with the damage also develop in the right part where the warp and wefts yarns are in directions close to ±45° (Figure 10) and, like the damage, this evolution is nonlinear since the strains double for an increase in force of 25%.

Figure 11 displays the non-linear nature of the force-displacement curve due to damage and inelastic strain, especially for imposed forces in excess of 2000 N.

## 5. Discussion and Conclusions

In this study, a simulation model for the prediction of yarns angle after forming process was coupled with a damage model for woven composite materials. The association of the two models demonstrated the importance of taking into account the change in angle between warp and weft yarns due to the manufacturing process when estimating the damage behaviour of composite structures. The forming simulation is based on a hypoelastic model for the textile reinforcement and it provides the yarn directions and therefore the warp and weft angle changes after the process. 

The analysis of the damage behaviour of the cured textile composite is carried out considering two UD equivalent plies in the yarns’ directions. It has thus been shown that this approach to the estimation of matrix damage makes it possible to take into account the reorientation of the fibres consequence of the forming process. The influence of this reorientation is very important and must be considered for a realistic analysis.

This study is part of the work carried out with the aim of taking into account the manufacturing processes when calculating composite structures. One of the first future developments concerns the extension of damage analysis to curved areas (single or double curvature) which are numerous in composite parts. In addition, experimental investigations are necessary to confirm and validate the influence of the manufacturing process on the damage.

## Figures and Tables

**Figure 1 materials-13-05337-f001:**
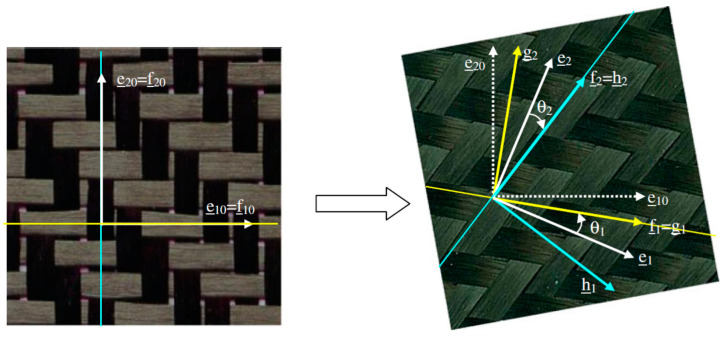
Orthogonal frames g (**g**_1_ = **f**_1_, **g**_2_) and h (**h**_1_, **h**_2_ = **f**_2_) defined from the direction of the warp and weft yarns.

**Figure 2 materials-13-05337-f002:**
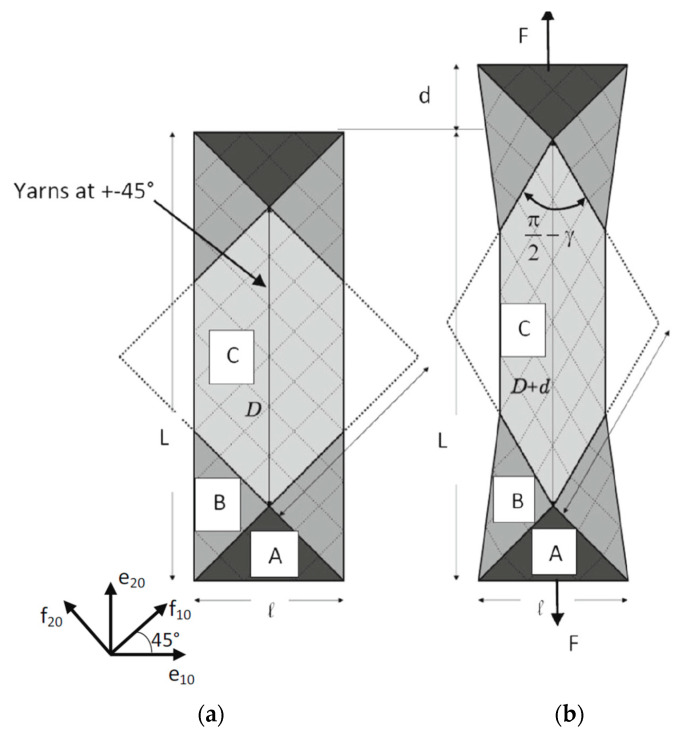
Bias extension test. (**a**) initial position; (**b**) development of sheared zone (C), half-sheared zone (B) and non-sheared zone (A); (**c**) finite element simulation and experimental thermoforming; (**d**) experiments.

**Figure 3 materials-13-05337-f003:**
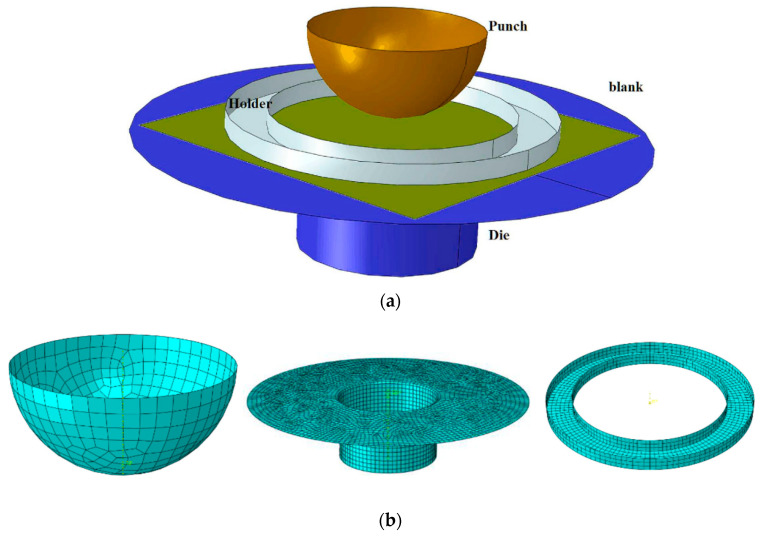
Hemispherical forming: (**a**) initial textile blank and (**b**) tools.

**Figure 4 materials-13-05337-f004:**
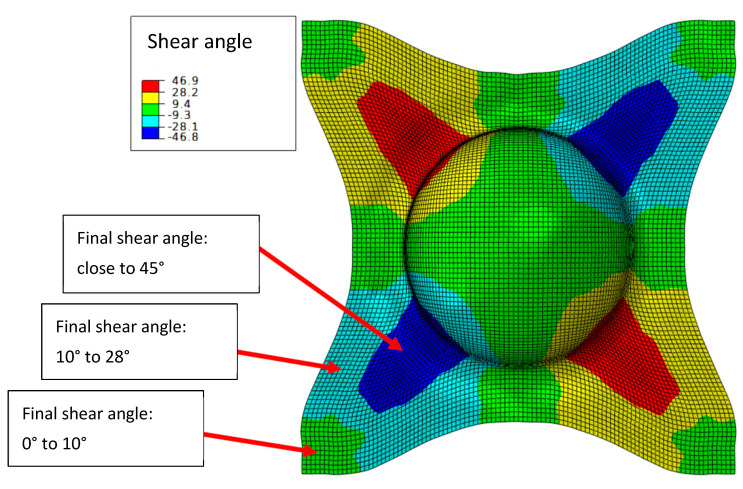
Shear angles after a hemispherical forming.

**Figure 5 materials-13-05337-f005:**
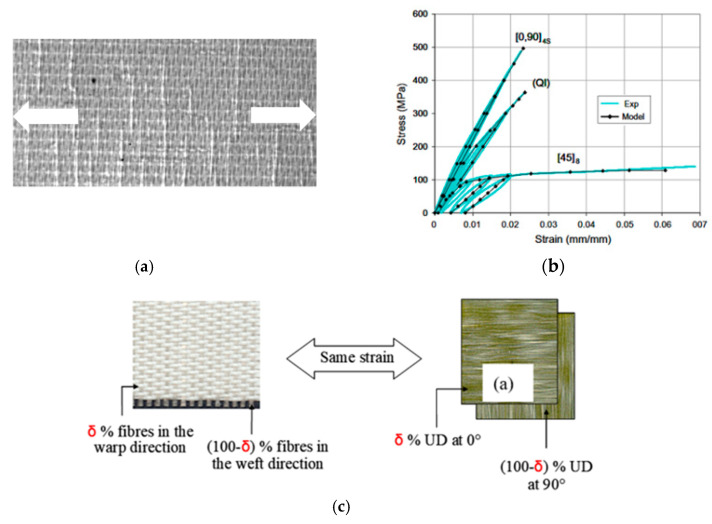
(**a**) Damage in a composite with woven reinforcement [23]. (**b**) Stresses versus strains in tension in different directions in experiments and model. (**c**) Assumption for woven plies.

**Figure 6 materials-13-05337-f006:**
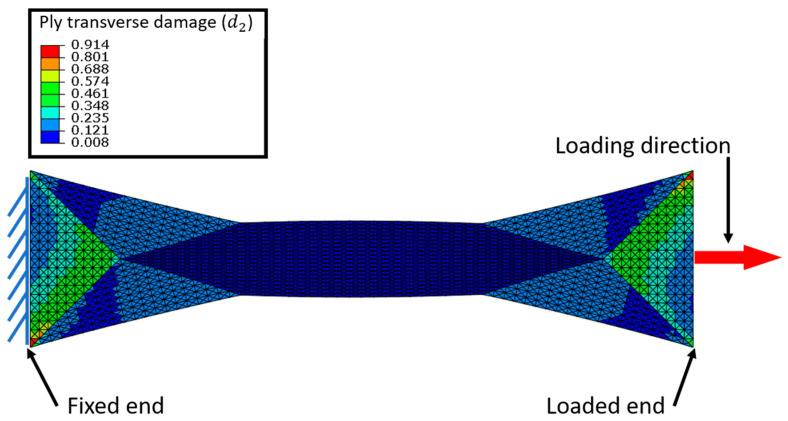
Boundaries condition for the simulation of tension test and simulated damage map.

**Figure 7 materials-13-05337-f007:**
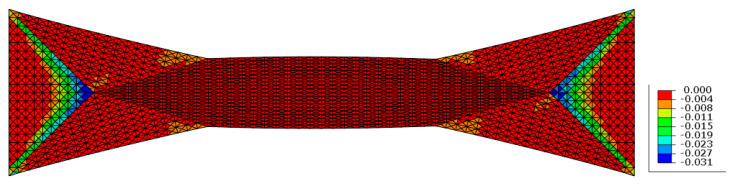
Inelastic shear strain of a specimen obtained via bias extension test after loading in tension.

**Figure 8 materials-13-05337-f008:**
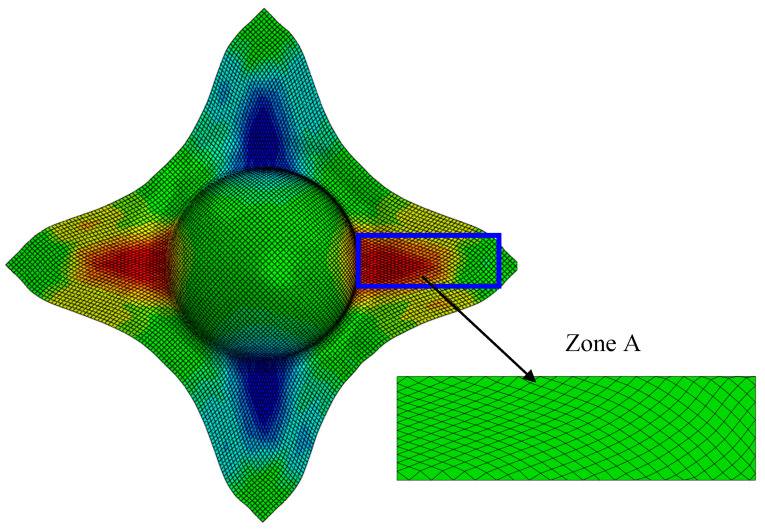
A specimen from the hemispherical forming (Zone A).

**Figure 9 materials-13-05337-f009:**
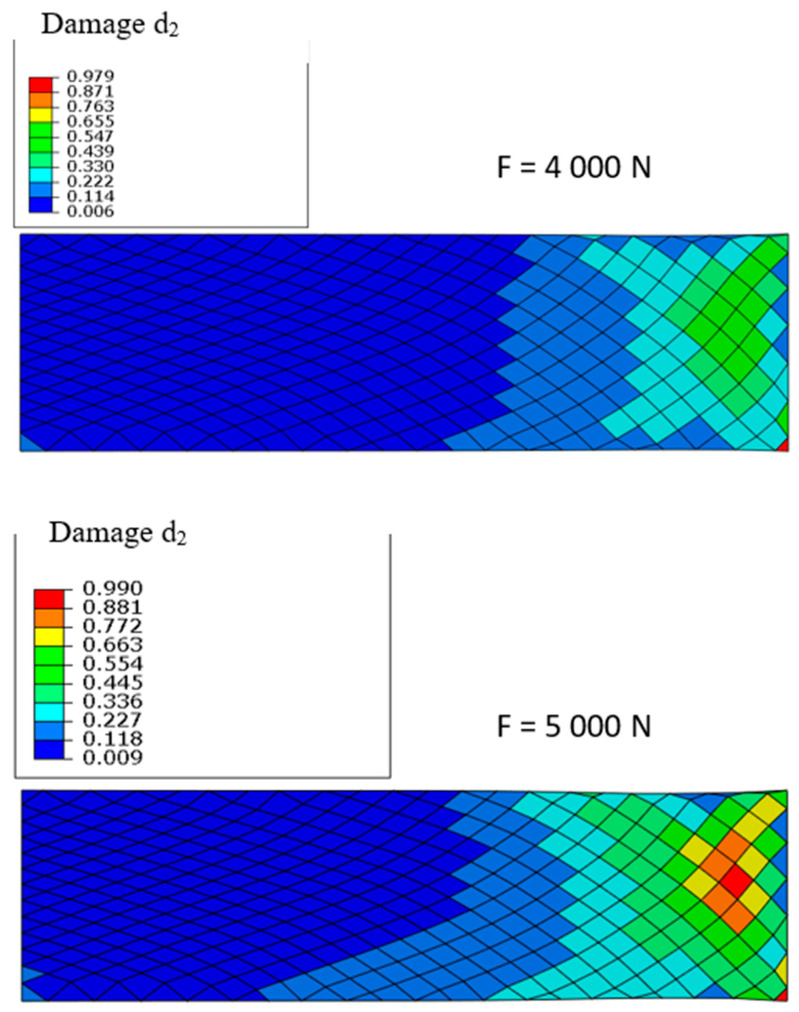
Matrix damage *d*_2_ for for 4000 and 5000 N.

**Figure 10 materials-13-05337-f010:**
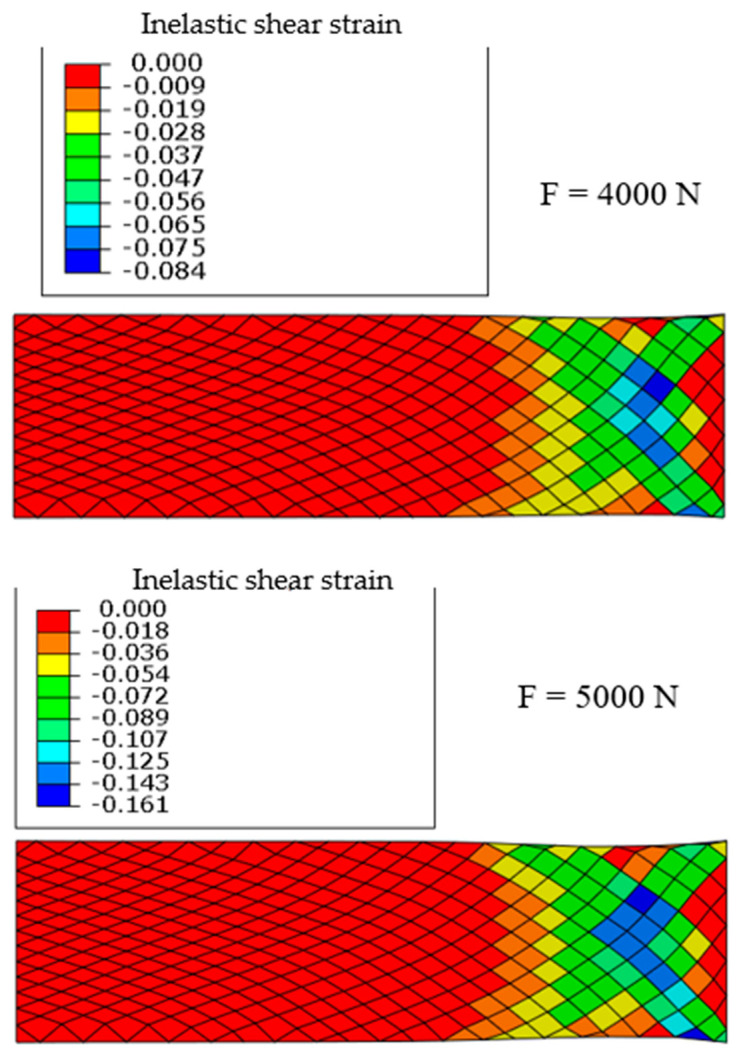
Inelastic shear strain for 4000 and 5000 N.

**Figure 11 materials-13-05337-f011:**
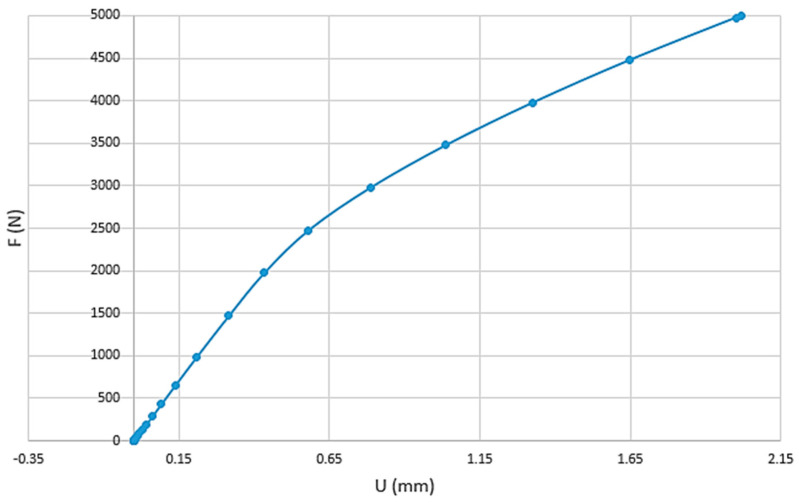
Non-linear force-displacement curve.

**Table 1 materials-13-05337-t001:** Mechanical properties of the textile reinforcement during forming.

Young’s modulus in warp direction E^g^:	35,400 MPa
Young’s modulus in weft direction E^h^:	35,400 MPa
Shear modulus G (the shear angle γ is in radians)$G=6.7135\gamma^{4}-9.8228\gamma^{3}+6.3822\gamma^{2}-1.5928\gamma +0.1948$	GPa

**Table 2 materials-13-05337-t002:** Damage properties of the cured textile composite [80].

Initial Young’s modulus in the fibre direction in a UD virtual ply:	$E_{1}^{0}$ = 40,000 MPa
Initial Young’s modulus in the transverse direction in a UD virtual ply:	$E_{2}^{0}$ = 13,000 MPa
Initial Shear modulus in a UD virtual ply:	$G_{12}^{0}$ = 4000 MPa
Poisson ratio of UD virtual ply:	$\nu_{12}^{}$ = 0.25
Parameters of the damage law:	a = 0.9, b = 0.32, c = 1, m = 0.75, n = 0.75, Y_0_ = 0 MPa
Inelastic strain in the shear direction	R_0_ = 60 MPa, h = 4500 MPa.

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
