# Peer review of "Textile Composite Damage Analysis Taking into Account the Forming Process"

_materials, 2020, doi:10.3390/ma13235337_

Round 1
Reviewer 1 Report
The article under the title: “Textile composite damage analysis taking into 2 account the forming process” is partly in line with the Materials journal. The authors work is focused on theoretical modelling. The organization of the article is appropriate, but it requires improvements:
- Abstract: required more details such as used materials and methods.
- Abstract: please give the full name “UD”.
- Introduction (line 25): sentence should started by capital letter.
- Introduction (line 26): please give the full name when you introduces first time the shortage.
- Introduction (line 31-32): please describe the most important information about the conclusion form these analysis.
- Introduction: please stress the novelty of the presented research.
- Forming process simulation (line 71): double bracket.
- Materials – please give the details about materials parameters, including input parameters for modelling.
- Figures 10 and 11 should be in part 3.
- Discussion – lack of comparison with literature or experimental data.
- Conclusion – add short conclusion part.
- COI – lack of information about COI.
- References enclosed a lot of positions, but a lot of them is older than 10 years..
- References: please verify the formats with the template.
Author Response
The authors wish to thank the reviewers for the useful comments and suggestions that were essential to improve the quality of our work. Response to the reviewer’s comments is given below.
In the revised manuscript, changes are highlighted with blue in the manuscript
Review 1
Comments and Suggestions for Authors
The article under the title: “Textile composite damage analysis taking into 2 account the forming process” is partly in line with the Materials journal. The authors work is focused on theoretical modelling. The organization of the article is appropriate, but it requires improvements:
- Abstract: required more details such as used materials and methods.
Author’s response:
The following details (in blue) concerning materials and methods have been added to the abstract, taking care not to exceed the 200-word limit.
Abstract: The internal structure of composite materials is modified during manufacturing. The forming of woven prepregs or dry preforms changes the angle between the warp and weft yarns. The damage behavior of the consolidated composite is modified by these changes of angle. It is important, when designing a composite part, to consider this modification when calculating the damage in order to achieve a correct dimensioning. A damage calculation approach of the consolidated textile composite that takes into account the change in orientation of the yarns due to forming is proposed. The angles after forming are determined by a simulation of the draping based on a hypoelastic behaviour of the woven fabric reinforcement. Two orthogonal frames based on the warp and weft directions of the textile reinforcement are used for the objective integration of stresses. Damage analysis of the cured woven composite with non-perpendicular warp and weft directions is achieved by replacing it with two equivalent Unidirectional (UD) plies representing the yarn directions. For each ply, a model based on Continuum Damage Mechanics (CDM) describes the progressive damage. Two examples are presented. A bias extension specimen and the hemispherical forming coupon. In both cases, the angles between the warp and weft yarns are changed. It is shown that the damage calculated by taking into account these angle changes is greatly modified.
- Abstract: please give the full name “UD”.
Author’s response:
Added in the abstract:
with two equivalent Unidirectional (UD) plies representing the yarn directions.
- Introduction (line 25): sentence should started by capital letter.
Author’s response:
A capital letter has been added at the beginning of the sentence line 25.
Both in the case of
- Introduction (line 26): please give the full name when you introduces first time the shortage.
Author’s response:
Added in the introduction (Line 32 of the revised version):
in the case of Liquid Composite Moulding processes (LCM)
- Introduction (line 31-32): please describe the most important information about the conclusion form these analysis. Please stress the novelty of the presented research.
Author’s response:
Added in the introduction (Line 52-54 of the revised version):
To our knowledge, there is a lack of a simulation method to calculate the damage in a loaded composite structure taking into account the effect of the forming of the composite. This is the objective of this article in the case of textile composites. It will also be shown that this taking into account of forming has important consequences on the calculated damage.
- Forming process simulation (line 71): double bracket.
Author’s response:
This mistake has been corrected
-Materials – please give the details about materials parameters, including input parameters for modelling.
Author’s response:
The mechanical properties of the textile reinforcement during forming (corresponding to the model developed in section 2.1) have been given in Table 1 added to section 2.1.
The Damage properties of the cured textile composite (corresponding to the damage model developed in section 3.2) have been given in Table 2 added to section 3.2.3
- Figures 10 and 11 should be in part 3.
Author’s response:
Figure 10 and 11 were moved to part 3.
- Discussion – lack of comparison with literature or experimental data.
Author’s response:
There is a lack of a simulation method to calculate the damage in a loaded composite structure taking into account the effect of the forming of the composite and comparison with literature is difficult. The aim of this work is to show the feasibility of combining a simulation of forming and a damage calculation on the consolidated composite part. The comparison with experimental results is one of the main perspectives.
- Conclusion – add short conclusion part.
Author’s response:
Added to ‘Discussion and conclusion’:
In this study, a simulation model for the prediction of yarns angle after forming process was coupled with a damage model for woven composite materials.. The association of the two models demonstrated the importance of taking into account the change in angle between warp and weft yarns due to the manufacturing processwhen estimating the damage behaviour of composite structures. The forming simulation is based on a hypoelastic model for the textile reinforcementand it provides the yarn directions and therefore the warp and weft angle changes after the process.
The analysis of the damage behaviour of the cured textile composite is carried out considering two UD equivalent plies in the yarns directions. It has thus been shown that this approach to the estimation of matrix damage makes it possible to take into account the reorientation of the fibers consequence of the forming process. The influence of this reorientation is very important and must be considered for a realistic analysis.
This study is part of the work carried out with the aim of taking into account the manufacturing processes when calculating composite structures.
- COI – lack of information about COI.
Author’s response:
Added at the end of the paper:
Conflicts of Interest: The authors declare no conflict of interest.
- References enclosed a lot of positions, but a lot of them is older than 10 years.
Author’s response:
Fifty percent of the references were published less than ten years ago. There are few or no references concerning the influence of forming on damage calculations. Nevertheless, articles on damage modelling of composites and to a lesser extent the simulation of forming are numerous and some are several decades old.
Reviewer 2 Report
The authors present a work dealing with woven composites and provide contributions on two fronts, namely a simulation of the forming process in two different situations (bias extension test and hemispherical forming), and a proposed simplified modeling approach on which the woven composite is modeled as a set of equivalent unidirectional plies. The topic is highly relevant, with high-fidelity modeling of fiber realignment processes in woven composites being an open issue on the development of reliable virtual mechanical testing tools for these materials.
The manuscript is well written and has a clear structure. However, I have a few points that I believe should be properly addressed before the work is accepted for publication:
- The proposed simplified model of using two equivalent UD plies to model the behavior of a textile layer is the main contribution of the work. Nevertheless, the authors do not give it enough space on the discussion presented in the manuscript. Other than a reference to two theses not written in English, no details are offered into how the simplified model is calibrated. Furthermore, it is unclear to the reader what is the magnitude of the modeling error incurred when opting for the simplified model, as comparisons with higher-fidelity models or experiments are never presented or mentioned. Since the remainder of the study hinges on the validity of this simplified approach, I believe a much more detailed discussion is necessary.
- The authors seem to employ continuum damage models without regularization, which would suggest the existence of severe mesh dependency and spurious snapback issues when using the approach on models experiencing global softening. However, no comments on the matter are offered in the manuscript. Did the authors experience mesh sensitivity issues? Did the authors test any of their models past the onset of global softening? How do the authors deal with the fact that at the limit of infinitely small finite elements the energy dissipation will tend towards zero?
- The discussion on the bias extension test is quite limited. How did the simulation results compare with the angles obtained from the actual experiment? Was the experiment used to calibrate the hypoelastic model which is later used to simulate hemispherical forming? Did the authors compare the final geometry of the hemispherical forming simulation with that of a real specimen?
- The damage profile of the bias extension specimen is shown in Figure 6, but it is never compared with that of a reference specimen with initial fiber orientation. I imagine damage distribution would considerably more uniform, but that would lead to another interesting point of discussion regarding what the authors expect will happen at the interfaces where damage behavior changes sharply.
I would also like to note a few minor points for improvement:
- Lines 135-137 and 137-139: These two sentences are redundant.
- Line 140: The word 'bias' appears twice.
Author Response
The authors wish to thank the reviewers for the useful comments and suggestions that were essential to improve the quality of our work. Response to the reviewer’s comments is given below.
In the revised manuscript, changes are highlighted with blue in the text
Review 2
Comments and Suggestions for Authors
The authors present a work dealing with woven composites and provide contributions on two fronts, namely a simulation of the forming process in two different situations (bias extension test and hemispherical forming), and a proposed simplified modeling approach on which the woven composite is modeled as a set of equivalent unidirectional plies. The topic is highly relevant, with high-fidelity modeling of fiber realignment processes in woven composites being an open issue on the development of reliable virtual mechanical testing tools for these materials.
The manuscript is well written and has a clear structure. However, I have a few points that I believe should be properly addressed before the work is accepted for publication:
- The proposed simplified model of using two equivalent UD plies to model the behavior of a textile layer is the main contribution of the work. Nevertheless, the authors do not give it enough space on the discussion presented in the manuscript. Other than a reference to two theses not written in English, no details are offered into how the simplified model is calibrated. Furthermore, it is unclear to the reader what is the magnitude of the modeling error incurred when opting for the simplified model, as comparisons with higher-fidelity models or experiments are never presented or mentioned. Since the remainder of the study hinges on the validity of this simplified approach, I believe a much more detailed discussion is necessary.
Author’s response:
. The simplified approach using two equivalent UD Plies has actually been developped in the two these [79,80]. It has been published and validated in Ref. [23] (Mechanics of Materials) . In order to present this more clearly the following paragraph has been added in section 3.1. Equivalent Unidirectional plies model of the woven ply
The identification of the material properties required by the model is detailed in [23,79] for an unbalanced woven glass/epoxy plies and static loads. The elastic and damage laws have been tested and validated for balanced woven carbon/epoxy in static [75,80] and extended for fatigue loads [77,78] plies. The proposed simplified model of using two equivalent UD plies to model the behavior of a textile layer has been used recently by other authors [81-83].
And in section 3.2.2 Damage evolution laws
The identification of parameters of the damage model presented above are given in details in [23]. The identification of the behavior typically requires 3 tests with loads and unloads: a test at 0 ° for the behavior in the fiber direction, a test at 90 ° for the transverse behavior and a test on a laminate [45,-45]ns for identify the shear behavior. The numerical values ​​of the parameters are given in the table 2. The paper [23] presents numerous validation tests on different laminates for static loads. The model has been extended to fatigue loads [77,78]. In this case, the matrix damage evolves during the cycles. Also in this case, the model has been identified and validated [77].
The part of the model presented here relates only to the nonlinear evolution of the matrix damage for static loadings. Failure of the laminate is not discussed here. This more complex part is presented in the papers cited. Rarely is the failure of the laminate due to matrix damage localization mechanisms because the laminate usually consists of plies whose fiber orientation corresponds to the direction of loading. On the other hand, the failure of the fibers is catastrophic for the laminate and the structure. The approach used in this case is described in [85] and takes into account the drop in resistance in the fiber direction as a function of the level of matrix damage and also takes into account the effects of better fiber strength in the presence of stress concentrations using non-local criteria [86, 87]. Let us note that these non-local criteria make it possible to regularize the problem and thus the dependence on the mesh for the numerical approaches.
- The authors seem to employ continuum damage models without regularization, which would suggest the existence of severe mesh dependency and spurious snapback issues when using the approach on models experiencing global softening. However, no comments on the matter are offered in the manuscript. Did the authors experience mesh sensitivity issues? Did the authors test any of their models past the onset of global softening? How do the authors deal with the fact that at the limit of infinitely small finite elements the energy dissipation will tend towards zero?
Author’s response:
Methods to avoid problems related to softening due to damage are not presented here. But the references of the papers presenting the methods that have been developed by the authors for this purpose are given in the following paragraph which has been added in section 3.2.2 Damage evolution laws
Rarely is the failure of the laminate due to matrix damage localization mechanisms because the laminate usually consists of plies whose fiber orientation corresponds to the direction of loading. On the other hand, the failure of the fibers is catastrophic for the laminate and the structure. The approach used in this case is described in [85] and takes into account the drop in resistance in the fiber direction as a function of the level of matrix damage and also takes into account the effects of better fiber strength in the presence of stress concentrations using non-local criteria [86, 87]. Let us note that these non-local criteria make it possible to regularize the problem and thus the dependence on the mesh for the numerical approaches.
- The discussion on the bias extension test is quite limited. How did the simulation results compare with the angles obtained from the actual experiment? Was the experiment used to calibrate the hypoelastic model which is later used to simulate hemispherical forming? Did the authors compare the final geometry of the hemispherical forming simulation with that of a real specimen?
Author’s response:
The bias extension test is an experiment that has given rise to numerous works and papers. For example the ref. [25-28] among many others. The Bias extension test gives two results. The first is kinematic and consists of the angles between warp and weft after deformation. The second is the force measurement, which allows determining the shear modulus of the textile reinforcement during forming. (It is usually the main goal of the test). In this study, it is the angles between warp and weft yarns that are important. They are taken into account in the damage calculation of sections 3 and 4. Nevertheless, if the test hypotheses are verified (inextensibility of the fibers and non-slippage at cross over), which is generally the case, the change in the angles between warp and weft are directly given by the displacement of the edges of the specimen.
Table 1 has been added. It specifies the mechanical properties of the reinforcement during forming, that are used in the hypoelastic model. The properties have been determined by tensile test, and picture frame test [54].
The simulation of a hemispheric preform forming and its comparison with experiment has been carried out in several studies. For instance Ref . [12, 44, 57]. The simulation are in good agreement with forming experiment. If the blank holder is axisymmetric, the shear angle are close to those obtained in Fig. 4.
- The damage profile of the bias extension specimen is shown in Figure 6, but it is never compared with that of a reference specimen with initial fiber orientation. I imagine damage distribution would considerably more uniform, but that would lead to another interesting point of discussion regarding what the authors expect will happen at the interfaces where damage behavior changes sharply.
Author’s response:
Indeed, the aim of this study is to show that when the orientation of the warp and weft yarns changes, the calculated damage is modifies. And that therefore angle changes due to forming must be taken into account. If a composite is tested where the angles are constant, the damage will be more uniform except at the edge. This can be seen in the result of the damage calculation on a tensile specimen after a bias extension test. Zones with a constant warp/weft angle lead to constant damage, possibly disturbed by the connections of the specimen.
I would also like to note a few minor points for improvement:
- Lines 135-137 and 137-139: These two sentences are redundant.
Author’s response:
The error has been corrected
- Line 140: The word 'bias' appears twice.
Author’s response:
The error has been corrected
the warp and weft yarn directions have been modified by the bias extension.
Reviewer 3 Report
The manuscript needs to be restructured so as to conclude the main findings. There is no conclusion section. The experimental results need to be clearly stated with data treatment using statistical parameters.
Author Response
The authors wish to thank the reviewers for the useful comments and suggestions that were essential to improve the quality of our work. Response to the reviewer’s comments is given below.
In the revised manuscript, changes are highlighted with blue in the manuscript
Review 3
The manuscript needs to be restructured so as to conclude the main findings. There is no conclusion section. The experimental results need to be clearly stated with data treatment using statistical parameters.
Author’s response:
Added to the discussion and conclusion section:
In this study, a simulation model for the prediction of yarns angle after forming process was coupled with a damage model for woven composite materials. The association of the two models demonstrated the importance of taking into account the change in angle between warp and weft yarns due to the manufacturing process when estimating the damage behaviour of composite structures. The forming simulation is based on a hypoelastic model for the textile reinforcementand it provides the yarn directions and therefore the warp and weft angle changes after the process.
Concerning experimental results, the aim of this work is to show the feasibility of combining a simulation of forming and a damage calculation on the consolidated composite part. The comparison with experimental results is one of the main perspectives. Nevertheless, the mechanical characteristics used for the forming simulations and for the damage calculations are given in Tables 1 and 2 respectively.
Reviewer 4 Report
Indeed, an interesting paper, results are presented and conclusions are made. It is acceptable after Major Revisions. My comments are below;
- What are the implications of this study? And why it’s important, the authors should explicitly mention in abstract and introduction.
- The authors should give strong literature argument for the need of this study.
- The authors should add more photos of real samples, for structural verifications, even though it’s a simulation work. What is LCM process? it should be defined, and the manuscript should be made easy to read for layman. English must be improved.
- The statement making this work interesting, simulation of warp and weft yarns angles, however could not portrait real experimental results. Is it possible to compare experimental results with your simulation work?
- Figure 2, show experimental sample, why we cannot see the experimental values? It should be provided in a table.
- The conclusion part should be improved and simplified focusing on main findings, rather than claims, a first step towards taking into account manufacturing processes in the calculation of composite damage”.
I enjoyed reading the paper, best wishes.
Author Response
The authors wish to thank the reviewers for the useful comments and suggestions that were essential to improve the quality of our work. Response to the reviewer’s comments is given below.
In the revised manuscript, changes are highlighted with blue in the text
Review 4
Indeed, an interesting paper, results are presented and conclusions are made. It is acceptable after Major Revisions. My comments are below;
Thank you for this comment
- What are the implications of this study? And why it’s important, the authors should explicitly mention in abstract and introduction.
Author’s response:
Added to the abstract:
It is important, when designing a composite part, to consider this modification when calculating the damage in order to achieve a correct dimensioning.
Added to the introduction:
To our knowledge, there is a lack of a simulation method to calculate the damage in a loaded composite structure taking into account the effect of the forming of the composite. This is the objective of this article in the case of textile composites. It will also be shown that this taking into account of forming has important consequences on the calculated damage.
It is important, when designing a composite part, to consider this modification when calculating the damage in order to achieve a correct dimensioning.
- The authors should give strong literature argument for the need of this study.
Author’s response:
In our opinion, there is very little literature concerning the consideration of forming in damage analysis.
Added in the introduction:
To our knowledge, there is a lack of a simulation method to calculate the damage in a loaded composite structure taking into account the effect of the forming of the composite. This is the objective of this article in the case of textile composites.
- The authors should add more photos of real samples, for structural verifications, even though it’s a simulation work. What is LCM process? it should be defined, and the manuscript should be made easy to read for layman. English must be improved.
Author’s response:
A photo that show damage in a composite with woven reinforcement has been added in Figure 5a. Indeed the study presented is a simulation work. He showed the possibility of combining a simulation of the forming process with a damage calculation of the cured composite. He also showed that the effect of forming on the damage calculation is important. The next step is to verify experimentally the results of the simulations.
It was clarified in the introduction that LCM processes are Liquid Composite Moulding process (resin in injected in a textile reinforcement).
The paper was corrected by a fluent English speaker.
- The statement making this work interesting, simulation of warp and weft yarns angles, however could not portrait real experimental results. Is it possible to compare experimental results with your simulation work?
Author’s response:
We intend to verify experimentally the results obtained. In particular, by testing a composite after a bias extension test which will modify the angles between warp and weft directions. The loss of stiffness due to damage can be compared to that given by the simulations.
- Figure 2, show experimental sample, why we cannot see the experimental values? It should be provided in a table.
Author’s response:
The bias extension test shown in Figure 2 gives two results. The first is kinematic and consists of the angles between warp and weft after deformation. The second is the force measurement, which allows determining the shear modulus of the textile reinforcement during forming. It is the angles between warp and weft that is important for the present study. It is taken into account in the damage calculation of sections 3 and 4. Table 1 has been added. It specifies the shear modulus.
- The conclusion part should be improved and simplified focusing on main findings, rather than claims, a first step towards taking into account manufacturing processes in the calculation of composite damage”.
Author’s response:
The conclusion have been modified:
In this study, a simulation model for the prediction of yarns angle after forming process was coupled with a damage model for woven composite materials. The association of the two models demonstrated the importance of taking into account the change in angle between warp and weft yarns due to the manufacturing process when estimating the damage behaviour of composite structures. The forming simulation is based on a hypoelastic model for the textile reinforcement and it provides the yarn directions and therefore the warp and weft angle changes after the process.
The analysis of the damage behaviour of the cured textile composite is carried out considering two UD equivalent plies in the yarns directions. It has thus been shown that this approach to the estimation of matrix damage makes it possible to take into account the reorientation of the fibers consequence of the forming process. The influence of this reorientation is very important and must be considered for a realistic analysis.
This study is part of the work carried out with the aim of taking into account the manufacturing processes when calculating composite structures.
I enjoyed reading the paper, best wishes.
Thank you for this comment.
Round 2
Reviewer 1 Report
The required changes have been implemented.
Author Response
The required changes have been implemented.
Thank you for this comment
Reviewer 2 Report
Most of the points raised have been addressed to some extent and references to relevant works have been added.
Author Response
Most of the points raised have been addressed to some extent and references to relevant works have been added.
Thank you for this comment
Reviewer 4 Report
The authors have modified the manuscript according to the given comments. But i do urge the authors should add real pictures with experimental data, even its a simulation work.
Author Response
i do urge the authors should add real pictures with experimental data, even its a simulation work.
A figure (5b) has been added that displays stresses versus strains curves in tension in different directions in experiments and model for a woven composite material.